# Sex-Gender Variable: Methodological Recommendations for Increasing Scientific Value of Clinical Studies

**DOI:** 10.3390/cells8050476

**Published:** 2019-05-17

**Authors:** Flavia Franconi, Ilaria Campesi, Delia Colombo, Paola Antonini

**Affiliations:** 1Laboratory of Sex-gender Medicine, National Institute of Biostructures and Biosystems, 07100 Sassari, Italy; franconi.flavia@gmail.com; 2Dipartimento di Scienze Biomediche, Università degli Studi di Sassari, 07100 Sassari, Italy; 3Value and Access Head, Novartis Italia, 21040 Origgio, Italy; delia.colombo@novartis.com; 4Nouscom SRL, 00128 Roma, Italy; thinkingabout807@gmail.com

**Keywords:** sex–gender, biomarkers, biorhythms, psycho-social and environmental factors, clinical trials

## Abstract

There is a clear sex–gender gap in the prevention and occurrence of diseases, and in the outcomes and treatments, which is relevant to women in the majority of cases. Attitudes concerning the enrollment of women in randomized clinical trials have changed over recent years. Despite this change, a gap still exists. This gap is linked to biological factors (sex) and psycho-social, cultural, and environmental factors (gender). These multidimensional, entangled, and interactive factors may influence the pharmacological response. Despite the fact that regulatory authorities recognize the importance of sex and gender, there is a paucity of research focusing on the racial/ethnic, socio-economic, psycho-social, and environmental factors that perpetuate disparities. Research and clinical practice must incorporate all of these factors to arrive at an intersectional and system-scenario perspective. We advocate for scientifically rigorous evaluations of the interplay between sex and gender as key factors in performing clinical trials, which are more adherent to real-life. This review proposes a set of 12 rules to improve clinical research for integrating sex–gender into clinical trials.

## 1. Introduction

### 1.1. Sex and Gender

At the beginning of this paper, it is important to make a distinction between sex and gender as there is some confusion as to the meanings of both words. Sex and gender are often used interchangeably, but it is important to point out that these terms should not be used as synonyms, especially when designing clinical trials. In general, sex refers to the biological, genetic (two chromosomes XX, as opposed to XY), and physiological processes related to sexual beings [1,2,3,4]. Whereas gender refers to the roles, relationships, and relative power that people relate to or societies generally attribute to women and men, irrespective of their genetic make-up. It is very difficult to separate sex and gender because they are multi-dimensional, entangled, and interactive [5,6]. Thus, in this paper, we will use the mixed term “sex–gender”, in order to recognize the value of both the biological and social contexts [5,7].

Progress in the physical, biological, social, and psychological sciences has been enormous. Unfortunately, psychological and social progress has not been effectively integrated into clinical medicine [8]. Thus, it is not surprising that, in the past, sex and gender have been neglected, and often viewed as annoying confounders [7,9,10,11,12,13]. Investigators have mainly studied only one sex, and this generates a lack of high-quality evidence ([12] and the references therein). Thus, physicians have to extrapolate medical recommendations from research mainly done on men, slowing down the overall implementation of sex–gender difference (SGD) detection in clinical research. Vice versa, the consideration of sex–gender and the assessment of their differences promote the health of women, men, and people of diverse gender identities of all ages, avoiding systematic errors that generate results with a low validity (if any) [7,13]. One systematic error was observed in the case of thalidomide. In the 1950s, thalidomide was used to treat nausea, or “morning sickness”, in pregnancy. Its use led to the death of about 2000 children and to serious birth defects in 10,000 children; even though pre-clinical tests in mice and rats did not show signs of teratogenic risk [14]. Successive testing in rabbits and other animals and in vitro testing on female human tissue predicted the toxicity of thalidomide [14,15,16,17]. These findings emphasized the pivotal role of species differences in drug effects. Additionally, the thalidomide tragedy changed the manner in which medications are tested.

Nowadays, at least among the academic community, awareness of sex–gender relevance in health and illness is present ([13] and the references therein). Aside from sex–specific organs, differences between men and women occur for a multitude of diseases and the related treatments [7,10,11,13]. Sex–gender impacts on pharmacological therapies have been considered with a relatively low relevance and, in general, male subjects are mainly used in research. Results are often extrapolated and thought to be applicable to women, which may not be true [12]. For scientific rigor, we must underline that the majority of studies that report on SGD did not have sufficient statistical power. In addition, the majority of manuscripts that we reviewed in writing this paper made no mention of their power for difference detection. There is also the possibility that studies that present similarities (negative papers), instead of differences, have not yet been published.

### 1.2. Pharmacological Response

The pharmacological response is multi-factorial, depending on drug (e.g., pharmaceutical form) and patient factors (genetics and epigenetics, age, body dimension and composition, metabolic processes, concurrent use of foods and drugs, including oral contraceptives (OC) and exposure to xenobiotics, lifestyles, ethnic, emotional factors, sex–gender, and so on) ([10] and the references therein). The analysis of many of these factors may present technical and/or ethical issues. Consequently, to catch the numerous interactions among medications, the human body, and the environment (past and present), it is necessary to overcome the reductionist approach of defining pharmacological responses only in biological terms [10,12].

In the past, and even in the present, the scarcity of women in randomized clinical trials (RCT) has been associated with the paucity of female animals in experimental pharmacology and with a lack of attention to the cellular level of sex differences in in vitro studies [12]. This leads to a scarce knowledge of aspects related to females. Instead, sex–gender can account for high (50%–75%) inter-individual variability in pharmacological responses [18]. The high inter-individual variability induced by sex–gender indicates that it should be adequately studied, to overcome the methodological and ethical issues and to reach a better applicability of the results to real-life [19]. Indeed, potential descriptive SGD in drug efficacy and safety profiles have been reported [7,10,11,20,21,22,23,24].

The SGD may have impacts on pharmacokinetics (PK), pharmacodynamics (PD), and safety profiles. The safety profile seems to be generally lower in women than in men [7,11,25,26,27]. Notably, most of the Food and Drug Administration (FDA) drug bans for toxic effects are guided by results from women [28]. However, some studies have indicated that the incidence of fatal and more serious drug adverse reactions (ADR) are more frequent in men than in women [29,30]. These studies are in line with the Swedish national pharmaco-vigilance database (2008–2011), which shows that women and men had higher rates of non-serious and severe reactions, respectively [31]. Currently, the main source of ADR for marketed drugs is represented by spontaneous reports and, evidently, this may determine a selection bias, with more severe ADR more likely to be reported. Importantly, women are more likely to report ADR [32]. Further, in general, women take more medication than men [33], and this arithmetically elevates their ADR risk and increases the risk of harmful pharmacological interactions. SGD in ADR probably originate from differences in PK and PD. Recently, in adult psoriatic patients, a significant difference in cyclosporine-induced ADR was presented between fertile and post-menopausal women, but not between women and age-matched men; suggesting the importance of hormones in drug safety [34].

Drug-induced cardiac arrhythmias represent an interesting case: men have, in fact, a greater risk of Brugada syndrome [35], whereas women present a higher risk for long QT-interval syndrome ([10] and the references therein). The case of painkillers is also relevant: tramadol, beyond nervous system toxicity, affects the male and female hearts differently. Tachycardia and S-wave abnormalities were over-represented in men, whereas long QT-intervals and right bundle branch blocks prevailed in women [36]. ADR may be recognized earlier in women if drug development trials include an adequate number of women and practice inclusion of the covariate sex–gender in the outcome analysis. Nowadays, these practices should be mandatory.

It is worth mentioning that Gartlehner et al., which reviewed numerous clinical studies that reported SGD in efficacy, found differences in efficacy in 1 out of 68 drugs, for 36 indications [25]. However, these results were compromised by some methodological limitations (for example, the study did not consider the adherence, or that women are more likely than men to be poorly adherent to prescribed medications ([10] and the references therein), and did not permit a clear evaluation of sex–gender effects on drug efficacy.

### 1.3. Sex–Gender in RCT

Until the early 1990s, testing drugs on women (particularly women of child-bearing age) was not recommended by the FDA [37]. Obviously, the exclusion of any participating population results in a lack of knowledge once the drug is put on the market. The FDA position was reverted in 1993 [38], and the new position was consequently implemented during the following years [39,40]. The FDA has produced several guidelines for the pharmaceutical industry ([41] and the references therein), which promote the collection and analysis of sex-specific data in clinical trials and dictate the demographic profiles of participants that should be included in FDA applications and that the results of clinical trials should be provided in terms of demographic indicators, such as age, sex, and race.

Still, there is a large preponderance of inadequate research on SGD. For multiple reasons, investigators are reluctant to shift the experimental paradigm, and women (especially women of child-bearing age) continue to be less-enrolled than men in the first, second, and third phases of clinical trials (Table 1) [42,43]. This is especially true in the early phases of clinical trials [44], although the number of women is slowly increasing [10,45]; for example, they are well-represented in RCT for arterial hypertension and pulmonary arterial hypertension [46].

In investigating SGD, investigators may encounter problems related to feasibility, logistics, methodological, and ethical issues when enrolling women. To overcome these issues, often the sex–gender influence is studied in sub-group analysis, but the power may be insufficient to adequately detect SGD ([41] and the references within). This generates controversial results, since most trials are not planned with appropriate sample sizes to detect treatment effects among sub-groups, or to have a precise estimation of the effects within the sub-groups.

Well-conducted RCT are considered the ‘gold standard’ for comparing different treatments. Ideally, RCT are performed with heterogeneous populations and provide average treatment effects. Participants (treated and control) are randomly allocated, by chance, to form groups that are supposed to be well-balanced for known and unknown factors. However, the neutrality of clinicians that include patients in the trial is not sufficient, if the variability depends on social groupings or other systematic differences (other than the intervention). In these cases, patient allocation may generate a bias, despite the randomization [47]; leading to both exaggeration and under-estimation of the results. To avoid or to reduce this bias, allocation concealment has been recommended [48]. Indeed, inadequate concealment of allocation may lead to a selection bias, which occurs when the selected sample is not representative of the patient population. This is especially relevant when enrollment is sequential and when recruiters can decide whether or not to enroll each eligible patient [48]. Entry criteria play a fundamental role, because they have the goal of ensuring that each individual in the trial is as similar as possible to any other, in order to ensure that treatment effect can be linked to the studied drug and not to other factors.

There is a low awareness of the importance of social and psycho-social factors on drug responses, and the drug discovery and development process still continues to use a ‘one size fits all’ approach. Consequentially, the participants of RCT may not be adequately representative of the population that will use the treatment [49]. RCT have a high internal validity, but there are concerns about their external validity (the extent to which results are generalizable), which can be often low [41]. Recently, other methods have received interest, in order to overcome the limitations of RCT [50].

The feasibility of a sex–gender balanced enrolment depends on the number of women in the studied population and on their characteristics (age, hormonal status, socio-economic status, personal autonomy, and so on), in relation to the study. If sex–gender is thought to be a modifier, leading to an interest in studying men and women, or sub-groups of men and women, separately, then an inclusive study would have a larger sample size (i.e., a sufficient size of each sub-group) than a study in which one or more of the sub-groups are excluded to achieve homogeneity, or in which a post-study subgroup analysis is carried out; even in the presence of a sex–gender stratified randomization. Finally, efforts should be made to attract women into RCT. For example, women may have less flexibility than men to keep appointments, especially when they must take care of children.

The endpoints of RCT can be classified in hard primary, such as mortality, and soft or intermediate endpoints that are usually based on surrogate parameters, such as biomarkers. Soft endpoints are permissive for a relatively small sample size; on the contrary, hard endpoints require larger trials, given the low frequency of the outcomes. The use of soft endpoints is preferred by sponsors. However, in sex–gender studies, it is necessary to know (if unknown) the influence of sex–gender on biomarkers (see below).

Globally, the consideration of sex–gender requires a re-thinking of the design and infrastructure of RCT, in order to ameliorate and offer more appropriate care to all people, and this is especially true in the era of the personalized medicine prevention and treatment strategies that take individual variability into account. Personalized medicine, in fact, should go beyond genetic information, including psycho-social and environmental data based on more reliable models for pre-clinical testing, statistically powered to detect treatment and sex–gender (or other) interactions [51]. Here, we suggest some rules to design trials with a sex–gender approach.

## 2. Rules

### 2.1. Rule 1: Trialists Should Define Terminology for Sex or Gender in Clinical Protocols

Currently, only few RCT define sex or gender for describing the demographic characteristics of populations. In the absence of a definition, the meaning of “sex or gender” varies among the studies, and they are sometimes used interchangeably. In light of the different conceptual constructs of sex and gender, it is relevant to define the two terms exactly to increase scientific value, and to better understand the influence of sex–gender on shaping health behaviors, opportunities, and outcomes [5,12,19,52] and cited literature. Therefore, whenever possible, both constructs should be investigated to understand their relative contributions toward the differences between men and women.

The second issue regarding terminology is the phases of the menstrual cycle (menstrual, follicular, ovulatory, and luteal), which are difficult to standardize. Different methods have been proposed, such as the actual day of the menstrual cycle, in standardizing to a 28-day cycle based on ovulation testing [53]. Indeed, the report of progesterone and estrogen plasma levels should facilitate comparison among trials and help to include women with irregular cycles. To ensure the enrollment of fertile women or women in menopause, follicle-stimulating hormone and luteinizing hormone should also be measured.

The third issue regarding terminology concerns the terms “estrogen, progestin, and androgen” to refer to very different patterns of steroid hormones. In fact, estrogen, progestin, and androgen refer to classes of hormones, but the single molecule may have totally different effects [54]. Thus, the specific hormone should be indicated. Overcoming these issues could facilitate comparisons among studies.

### 2.2. Rule 2: Research Teams Should be Trained to Avoid Investigator Sex–Gender Bias

Sex–gender research often does not consider the researcher, forgetting that he/she is a person and that every individual is sexed and gendered ([12] and the references within). However, the results of the research, practices, and attitudes of physicians may depend on their own sex–gender [55,56,57,58]. In particular, men and women tolerated experimental pain better when the care providers were of the opposite sex–gender [59], or was a woman [60]. Men reported lower pain to female physicians versus to male ones, and this happened in the absence of significant changes in heart rate and skin conductance, suggesting the importance of psycho-social factors [61]. Female care providers recommend more psycho-social treatments for pain in women than in men [62]. Recently, a significant difference in the performance of key elements of cardiac examination was shown. Male residents were less likely than female residents to correctly perform each maneuver on a female (versus male) standard patient with chest pain [63]. Further confirmation comes from the paper [12], which examined patients with lower back pain and showed that an observer was significantly influenced by patient sex–gender and by their physical attractiveness [64]. In addition, it has been described that male and female researchers focus on different aspects of behavior and interpret the same results in very different ways [65]. Both health-care providers and patients are influenced by their respective cultures, and the influence of culture on health and disease is of the greatest relevance. Nowadays, cultural diversity is increasing globally, affecting perceptions of health, disease and death, prevention and health promotion, pain experiences, preference for type of treatment, contraception, and so on. Clinical researchers should overcome ethno-centrism and should have a cultural competency, in order to obtain the required information from patients of different cultural backgrounds, especially when research and clinical practice occur in a gender perspective [66]. A clear example is the influence of religion and culture on sexual and reproductive behavior and health [67].

The diversity observed in male and female care providers could be a non-defined (but potentially important) source of sex–gender bias in clinical trials, where the investigator/health-care provider may modify the allocation and medical outcomes [68]. Further, the patient–investigator interaction may also play a role in placebo and nocebo responses, considering that most of the placebo effects arises from the psycho-social setting in which a treatment occurs (see below).

In conclusion, a research team should be constituted of women and men with cultural competency, and the leadership should include men and women, in order to reduce or to avoid inappropriate conclusions linked to the sex–gender of the researchers; furthermore, they should be trained to avoid investigator sex–gender bias.

### 2.3. Rule 3: Appropriate Inclusion of Sex–Gender as a Basic Variable in RCT Should Consider the Whole Human Life-Span

The demonstration of true SGD or sex–gender similarities after administration of a medication requires the consideration of some issues, and one of the most important is age [9]. Sexual differentiation starts during pre-natal life and continues throughout the life-span [69]. Relevantly, drug metabolism is influenced by sex–gender and age [70], as well as the activity of some inotropic and metabotropic receptors [71].

Finally, women and men can develop diseases at different ages [13]. This last fact limits the use of arbitrary age cutoffs in the inclusion criteria of RCT. Further, men and women over 65 years have specific sex–gender-associated health problems. For example, older women use more drugs and are subject to more ADR [72,73]. Further, chronic social isolation prevails in old women, which increases the risks of morbidity and mortality [74].

Consequentially, well-designed studies should enroll a sufficient number of women and men of different classes of age, and the sex–gender-age-associated differences should be considered in advance, including specific controls to prevent false-positive or -negative results. Indeed, exclusion criteria based on comorbidities and polypharmacy could limit the potential participants to a greater extent for women [75].

### 2.4. Rule 4: Patient History-Awareness of Socio-Economic Status, Education Levels, Stressors, Microbiota, and Geographical Localization, as well as Specific Sex–Gender Aspects

Most RCT include little data on psycho-social and demographic characteristics, which reduces the applicability of results from RCT to specific populations [76]. Indeed, the history of patients should include considerations of their function in daily life, such as productivity, social roles, intellectual capacity, and emotional stability and well-being [77]. Recently, Fava et al. created a road-map for integrating bio-psycho-social data, in order to build a detailed characterization of each individual enrolled in an RCT [77]. In other words, there is a need for multi-dimensional patient histories, keeping in mind the unity of biological (sex) and psycho-social aspects (gender) (Table 2). In fact, disparities in culture, education, lifestyles, stressors, environmental conditions, social status, loneliness, and access to care and health-care systems create different scenarios that may modify drug responses [10,78,79]. Consistently, social status, violence, and caregiver roles are related to the stress response, increasing morbidity and mortality ([12,80] and the references within).

Rating scales for psycho-social aspects are available and are largely applied, but many intellectual and economic barriers limit their use in RCT. However, the implementation of “clinical pharmaco-psychology and social pharmacology“ are mandatory in sex–gender studies, because their rating methods would help to catch all of the complex interactions of factors that lead to drug responses in men and women.

Particular attention should be dedicated to the place where one lives, or where one has lived. In Europe, the impacts of overweightness/obesity, smoking, and educational differences on cardiovascular diseases are country and sex–gender specific [81]. Additionally, the activity of some drugs appear to be country-specific. For example, SGD in survival with zofenopril is smaller among people living in Northern Europe and larger in people residing in Southern Europe [82]. This is of special relevance for multi-national studies, where the variability induced by the country (or by geographical region) should be considered. Multiregional trials, which have the advantage of being able to identify world-wide efficacy and safety of medications, because of their small sample size in the single country and/or geographical region, do not permit individuation of the variability that may vary across countries [83,84].

Attention should also be paid to the gut microbiome, because it is affected by socio-economic status [85], by foods, and by pregnancy and lactation [86]. It is also modified by drugs, such as antimicrobials and anti-cancer drugs, in a sex–gender specific way, both in humans and in animals ([12,87,88] and the references within).

In addition, patient history should pay attention to specific factors for a single sex–gender, such as: polycystic ovary syndrome, age at menarche, and menopause without forgetting pregnancy, and eventually the complications, such as gestational diabetes and hypertensive disorders, that can place a woman at long-term risk for the development of cardiovascular diseases. The use of certain anti-cancer drugs and radiation therapy for breast cancer, the current and the past use of OC, and use of hormonal replacement therapy (HRT) should be known.

Consequently, a detailed patient history is needed for sex–gender studies, in order to understand the interactions among biological aspects (age, genetics, hormones, and so on) and environmental ones, to effectively design personalized medicine and to ameliorate the appropriateness.

### 2.5. Rule 5: Detection of SGD in Placebo and Nocebo Responses

Placebo and nocebo effects and responses are present in every medical intervention [89]. The placebo and nocebo responses are influenced by physiological processes that create placebo and nocebo effects [90], and by other numerous factors (living conditions, illness characteristics, previous treatments, self-management, treatment setting, and patient conditions) [91]. In 2007 [9], one of the authors hypothesized that the placebo response could be different in men and in women. A recent systemic review (identifying 12 and 6 studies on placebo and on nocebo effects, respectively) showed that men and women respond more strongly to placebo and to nocebo, respectively [92]. In addition, this review made it evident that different stimuli determine placebo in men (verbal information) and nocebo in women (conditioning procedures). On the contrary, only 3 of 75 studies included in another meta-analysis detected an SGD in placebo response [93].

The magnitude of the placebo response is highly variable, both within and across therapeutic areas, and the influence of sex–gender would have a paramount consequence in the evaluation of results from RCT. Trial methodologists and regulators routinely demand, and researchers routinely attempt, to include placebo groups in every major RCT. In line with this requirement, it could be useful to select study participants based on pre-treatment response to placebo in attempting to evaluate placebo response in men and women.

### 2.6. Rule 6: PK of the Excipient and Active Drug According to a Single Route of Administration, Hormonal Fluctuations, and Alcohol and Tobacco Use

Many human PK studies have been conducted in a small number of individuals (usually healthy adult men) [94], which compromises the ability to detect small differences. Conversely, when differences are detected, one wonders how much credence to place in the finding [95]. To stress the relevance and importance of SGD in PK, we wish to report two examples. Women treated with metoprolol have higher plasma levels of the drug and more prolonged hypotension and bradycardia than men [96]. However, dose adjustments are not requested in the regulatory approved drug information. To reduce the dosage of zolpidem in women, we had to connect the association of zolpidem use with more than 700 motor vehicle crashes [97].

SGD may also arise from the administration route (intra-venous, oral, or intramuscular) [12]. This point is often neglected, despite the fact that dosing regimen, dosage form, formulation (e.g., composition or makeup of the dosage form) may all be influenced by sex–gender ([12] and the references within). Another complicating factor is the fasting (or not fasting) status, and differences in gastro-intestinal pH and mobility and gastro-intestinal transit time (Table 3).

The PK differences could be ascribed to active drugs and/or excipients. The excipient polyethylene glycol 400 enhanced the bioavailability of ranitidine only in healthy men [124]. This is very relevant in equivalence studies, which are mainly performed in healthy men [125]. Nevertheless, regulatory authorities dictate that they should be performed in men and women in similar proportions, because sex-based analysis promotes the efficacy and safety of generic drugs [126]. Thus, regulatory authorities should adopt a more precise, strong dictate.

Beyond excipients, SGD in PK also involves a multitude of active principles, including biological ones [9,10,11,98,127,128,129,130] and cited literature]. Currently, a fixed dose (estimated for young adult man of 70 Kg) is used in the majority of cases, independent of sex. However, when 10 mg of medication is adjusted for body weight, a typical adult Caucasian man and woman receive 700 and 570 mg, respectively. If the dose is adjusted for body surface area, a typical adult Caucasian man and woman receive an average dose of 703 and 608 mg, respectively [131].

In women, the PK is also influenced by changes in the hormonal environment, which occur during the menstrual cycle, pregnancy, menopause, hormonal treatments (such as OC, HRT), and other drugs, which influence hormonal status. For example, during ovulation, the absorption of aspirin and alcohol is decreased [53], whereas the dose of antipsychotics might need to be raised in pre-menstrual phases [132]. However, in humans, it may not always be important to test at different phases of the menstrual cycle, but it is relevant to have in mind whether such testing is appropriate. The use of exogenous sexual hormones may modify drug clearance, mainly through the inhibition or induction of cytochrome P450 enzymes (CYP) ([9] and the references within). Considering that, during RCT, contraception is requested for women of child-bearing age, this point could be very relevant.

It is well-known that metabolism presents numerous SGD (Table 3), which can vary in pregnancy, menopause, aging [133]. Psycho-physiological stress can influence drug-metabolizing systems, with potential consequences on drug therapy [134], and this could occur in a sex–gender-specific way [10].

Lifestyle choices (such as smoking and alcohol use), association with other drugs (including herbal and over-the-counter products), diet, and exposure to pollutants, genetic and demographic factors, and diseases may modify PK [135], and this could also occur in a sex–gender-specific way [10]. However, sex–gender does not influence the PK of all drugs, and many PK differences are less than 40% [136].

Currently, during the drug development process, PK/PD analysis is often performed during phases 2 or 3, or even when they are completed [94]; it will be relevant to prevent drug development from entering phase 2 and 3 trials without the necessary PK. Afterwards, one could evaluate data from approximately nine women, using Bayesian analysis methods. If the data of the women and of the men are similar, it could be assumed that there are no significant PK, and researchers could go into phase 2 and 3 trials using the same dose in both men and women, or using a drug dose normalized for body weight, body max index, or body surface area [94]. If the data diverge between men and women, a further study of approximately 19 women may be required to determine the appropriate dosing for women [94]. It would be appropriate to perform PK during phase 2, with the dose finding in men and women considering fertile and un-fertile ages, at least for women.

Women are un-fertile in the post-menopausal age, and illnesses increase dramatically in this age; thus, excluding menopausal women makes little sense. The use of a physiologically-based pharmacokinetic approach (PBPK) can facilitate the investigation of SGD by predicting biological differences in all PK parameters [137]. However, PBPK may not detect the gender effect. Establishment of appropriate dosage regimens requires knowledge of PK in the target population.

### 2.7. Rule 7: The Exclusion of Women of Child-Bearing Age Does Not Permit the Appropriate Pharmacological Therapy

The recruitment of women of child-bearing age in RCT involves sensitive ethical issues, such as the use of contraception. Contraception is usually requested to avoid damage to the fetus, but contraception practices are greatly variable (such as its governance). However, this is very difficult, as international regulatory guidelines on birth control requirements in RCT are not harmonized [138], and it also depends on the religious beliefs of the individual woman and/or its family, who must have the option to select anti-conception methods [67]. The use of only one type of contraception (or of the same OC) would increase the scientific rigor of the trial, even if, the intake of exogenous hormones could introduce more confounders.

Hormonal contraception (HC) may have different effects when compared with barrier methods. HC affects many clinical chemistry and physiology variables, as well as DNA methylation, endothelial function, macrophage function, and immune responses; thereby, compromising the studies intended to represent all women [139]. Finally, a single OC may have different effects. For example, the use of androgenic and non-androgenic progestin may differently affect hematic, biochemical (including DNA methylation), and macrophage parameters (including the expression and activity of estrogen receptors) [139,140].

Changes induced by OC may affect both PK and PD [141]. Further, OC may produce ADR that may alter patient compliance and adherence to test treatment and may, also, interact with the test medication. In particular, OC effectiveness can be reduced by the tested drug and, vice versa, the efficacy and safety profile of the test drug can be altered by OC [9]. Therefore, the specific risk of a single study relative to either a real or potential pregnancy should be carefully considered for men and women based on scientific data.

Informed consent, including parental consent (if possible), should be carefully prepared and should offer the opportunity to become aware of the issues linked to the specific form of birth control. The anticipation of reproductive toxicology (which may not be completed before the beginning of phase 3) and of pre-and post-natal development studies, which are completed even later [142], could encourage the enrollment of women of child-bearing age. If the studied molecule has teratogen effects in an animal study or early-phase trial, the participants should avoid becoming pregnant during the study.

According to the American College of Obstetricians and Gynecologists [143], the requirement for contraception should be tailored for specific study design and should be based on the actual risks to the pregnancy of an individual research participant.

### 2.8. Rule 8: Clinical Studies in Pregnancy and Lactation

Due to concerns about the safety of the fetus, only a few medications have been tested in pregnancy. The FDA has encouraged clinical studies in pregnant women, paying attention to the potential fetal risk more recently [144,145]. The enrollment of pregnant women has had some success, considering that, between 2009 and 2011, Endicott and Haas identified 264 drug trials that had enrolled pregnant women, including 23 PK studies and 130 placebo-controlled trials [146]. Nevertheless, the pregnant women enrolled represented 1.29% of the total registered trials from the 1960s through to 2013, and many of them (97.7%) did not have sufficient information to determine teratogenic risk [147]. In 2005, the European Medicine Agency recognized the need for more information on drugs in pregnancy and proposed active post-authorization surveillance of either new or old drugs [148]. Usually, post-marketing monitoring uses passive mechanisms to detect ADR and this presents some limitations, which can be reduced if pregnancy registries are prospective studies. The registries also have the advantage of overcoming ethical issues, although they still present some limitations (deficiencies in participation, false positive in outcomes, selection bias, and loss of patients in the follow-up) [147]. In 2009, the FDA embarked on a systematic study of the outcomes of prescription drug use in pregnant women [148] and in 2018, the FDA [145] prepared a guidance for industry for the inclusion of pregnant women in clinical trials.

It is evident testing drugs during pregnancy presents a challenge due to the presence of many factors (including the ethical ones) and the reluctance to expose the fetus to medications; although, there is evidence that pregnant women are showing an increasing willingness to participate in drug research [149]. The difficulty of including pregnant and breastfeeding women in RCT could be helped with the use of big data, collected through electronic medical systems, which usually refers to daily routine clinical practice. The use of big data is of value but does not have the scientific rigor of RCT [147].

Certainly, women and healthcare providers need more information on the effects of drugs when they are used in pregnancy and lactation. This is a public health issue since: 1) there is a widespread use of prescription drugs during pregnancy and at least 50% of pregnant women use one or more over-the-counter medications during pregnancy ([147] and the references within); 2) many women have medical conditions when they become pregnant or become ill during pregnancy. Age is of particular importance, at least in developed countries, because the age of pregnancy is gradually increasing [150] and, with age, the prevalence of diseases elevates dramatically [151]. Notably the reluctance of the care provider and of the patient to treat medical conditions may lead to the paradox of increasing the relative harm to the woman and the fetus than if she had been treated; 3) only rarely have drugs been formally or adequately investigated during pregnancy ( [147] and the references within); 4) the body of a woman changes throughout pregnancy (Table 2), involving almost all organs and systems, which may induce profound alterations in the PD and PK of many drugs [99], impacting the achievement of therapeutic levels [152].

In addition, both the placenta and the fetus can contribute to drug PK [152]; however, their effects are small when compared with maternal effects on the drug. Importantly, the body changes depend on pregnancy timing [99], and some pregnancy databases suggest that the teratogenic effects of drugs, such as valproic acid, carbamazepine, phenobarbital, lamotrigine, topiramate, and lithium, may be dose-dependent [99]. The above considerations may lead to specific dosage schedules, not only in pregnancy but at specific times during pregnancy ([147] and the references within).

The exclusion of pregnant and breastfeeding women in RCT precludes the attainment of knowledge regarding how drugs work in pregnant women and complications, which may not be known at the time of drug approval [153]. Even after the dramatic examples of diethylstilbestrol and thalidomide, in our opinion, it is urgent to increase the knowledge on drug effects in pregnant women through the use of multi-disciplinary research teams (obstetrics, pharmacologists, and so on); furthermore, it should be considered relative to the sex–gender of fetus. As sustained by the FDA, the inclusion of pregnant women in RCT is necessary to keep in mind that are many maternal and infant confounders [154]. Among infant confounders, it is necessary to recall the sex–gender of the fetus [155]. For example, maternal meclozine reduces the risk for hip subluxation in girls, but not (or at least less) in boys [154]. Among maternal confounders, we also recall age, alcohol use, body mass index, social economic status, and so on. Socio-economic status could be of relevance in societies with large socio-economic differences, as compared to welfare societies [154].

To include pregnant and breastfeeding women in RCT requires the understanding of the differences in the PK and, as already mentioned, the anticipation of reproductive toxicology studies and pre-and post-natal developmental studies would be beneficial [142]. The construction of a PK–PD model, measuring drug levels in parallel with medication effects, would help to reach evidence-based decisions in dose schedules during gestation [137]. Some of these models are now available [156], but they do not incorporate gender aspects.

Although serious acute ADR from drugs in breast milk appear to be uncommon. Measurement of drug concentrations in humans remains the golden standard. The new FDA requirements should improve the use of medications in nursing mothers ([157] and the references therein).

Finally, a major point to be addressed is the length of follow-up for infants born from pregnant and breastfeeding women enrolled in clinical trials in consideration of epigenetic modifications that some drugs may have induced and that drug induced epigenetic modifications may be influenced by sex–gender [158].

### 2.9. Rule 9: To Be Aware about the Influence of Sex–Gender on Biomarkers

The efficacy of drugs should be tested in studies, which assess final patient-relevant outcomes. Market access is often based on surrogate endpoints that use biomarkers, which replace and predict patient-relevant outcomes unavailable at the time that the new compound is put on the market [159]. Biomarkers can objectively measure and evaluate physiological and pathological processes, allowing an estimation of patient responsiveness to specific therapies.

Moreover, they are widely used as primary endpoints in RCT. For example, cholesterol, HbA1C, and viral load for human immunodeficiency virus are widely used as endpoints to obtain regulatory approval for cardiovascular diseases, diabetes, and acquired immune deficiency syndrome [160]. Biomarkers may lack reproducibility, potentially due to the variations induced by biological and lifestyle factors, which are not taken into account in the study design and statistical analysis [161].

Human biomarker levels are influenced by numerous factors, such as time of sampling and time of storage, food intake before sampling, urbanity, age, smoking status, geographical location, climate, sex–gender, and so on ([162,163] and cited literature). In biological fluids, the levels of numerous molecules (Table 1) show significant differences between men and women [109,122,133,164,165,166,167,168,169,170,171,172].

Biomarkers are also influenced by hormonal status, making it important to identify women in the follicular or luteal phases of the menstrual cycle, users of OC, and post-menopausal women [163]. Additionally, exogenous hormones (HRT and OC) influence biomarkers [122,164]. A recent paper, which measured 171 serum proteins and small molecules, showed that 96 molecules varied with sex and that 66 molecules varied between OC users, post-menopausal females, and during the menstrual cycle [164]. This last aspect could be of the largest relevance in RCT, where women of fertile age use OC for birth control.

Body dimension and composition, which differ between men and women (Table 3) have an impact on biomarkers, attenuating or even completely reversing the sex–gender associations after adjusting for these variables [109,120,169].

As already mentioned, the levels of biomarkers vary during the day. Notably, the survival in men with metastatic colorectal cancer is prolonged by 3.3 months with chrono-administration of oxaliplatin, 5-fluorouracil, and leucovorin, versus conventional delivery [173]. Further, human chronotype depends on age. Considering that basal individual biomarkers may be affected by sex–gender, we suggest that the selected biomarker/s should be analyzed, in advance, in both sex–genders, to prevent false-positive or -negative results when used as selection criteria and/or endpoints in RCT.

### 2.10. Rule 10: Lack of Detection of SGD Must Be Reported

Even if SGD are not present after the sex–gender analysis, trialists and researchers should, nonetheless, report the absence of SGD as a critical and relevant piece of scientific information.

### 2.11. Rule 11: Ethics Committees Need to Put on Sex–Gender Glasses

Research ethics committees ensure that participants are protected from harm, but they should also have a role in the implementation of sex–gender attention in evaluating whether (if necessary) there is equitable representation across the sex–genders. In order to obtain this result, it is relevant to enhance the knowledge of members of ethic committees on how sex–gender issues can be included in the design and conduct of clinical studies and the ethical assessment of study protocols. Among components of ethical committees, the presence of sex–gender experts with cultural competences should be mandatory to ensure that studies have been designed so as to consider all the individuals who are to be treated.

### 2.12. Rule 12: The Creation of Gender Alliance

The inclusion of women may require a large number of subjects because small samples will be insufficiently powered to detect interaction effects [174]. Most importantly, there is an opportunity to recruit women and men separately, so that the study intervention can be examined both within and across these groups and to assess interactions of social determinants with other variables, to increase the accuracy and meaning [175].

It is clear that the inclusion of women in RCT increases the complexity, research time, cost, and space used for research. However, these are not valid reasons to avoid the implementation of women [23], as only with their inclusion is possible to offer more appropriate care and to reduce individual and social costs. Hence, it is time to sustain that the SGDs are not “yes” or “no”, but that they lie between yes and no [176].

It is evident that sex–gender-sensitive biomedical research requires an alliance among the pharmaceutical industry, regulatory authorities, the health-care system and providers, researchers, and patients. In this alliance, scientific journals play a crucial role, as they should require sex–gender analyses of results, in order to avoid scientific discoveries continuing to be hampered by the under-investigation and under-reporting of sex–gender in pre-clinical and clinical research. In our opinion, a pre-planned analysis of outcomes and a sufficient statistical power for sex–gender should be required.

Thus, it is urgent to develop an integrative system to carry out research within these complex landscapes, which should encompass the presence of different competencies and of different sex–genders in the team that designs the study protocol ([12] and the references therein).

## 3. Conclusions

The awareness of SGD and similarities in prevention, diseases, and treatments is a relevant step towards reconciling clinical practice and personalized medicine in the real world. Thus, to maximize the scientific rigor and value of the research, it is mandatory to make a concerted effort to ensure that sex–gender-specific analyses are included in both pre-clinical and clinical research, in order to ensure health equity and to ameliorate the health and well-being of all citizens. All countries should ensure that health and regulatory agencies promote sex–gender awareness and prioritize sex–gender research, evaluating sex–gender differences and similarities over the human life-span, including pre-natal life. We are more prone to accept that differences mainly depend on genetic or hormonal influences, rather than being due to experience; however, sex–gender acts as an independent variable that involves many other variables.

Therefore, there is a need for new and reliable study design and modeling focused on the biological differences between men and women. RCT should also include designs and models that integrate and consider the impact of socio-cultural conditions on their results and outcomes because it is not possible to have efficient and safe therapy for all without the consideration of psycho-social, cultural, and economic factors. Notably, psycho-social factors can affect men and women differently and, thus, there is a need for investigations focused on sex-by-environment studies.

Here, we present some issues regarding the design of clinical trials and the analysis of the resulting data to accurately and efficiently detect sex–gender differences and similarities.

For old drugs, it would help to use the “bedside to bench” approach, instead of the classical “bench to bedside” approach, in order to identify patient-oriented selection strategies [177] through the identification of SGD in clinical outcomes to conduct further patient-oriented research.

## Figures and Tables

**Table 1 cells-08-00476-t001:** Definition of clinical studies.

Phase	Definition
0	This phase, also called human micro-dosing studies, includes the administration of single sub-therapeutic doses of the studied drug to a small number of healthy subjects (10 to 15), to gather preliminary data on pharmacokinetics (PK).
1	This phase tests side effects, maximum tolerated dose and the dose-limiting toxicity, and drug formulation in a small number (20–100) of (often healthy) individuals.
2	This phase assesses the preliminary clinical safety and efficacy of selected doses in a dozen to a hundred patients with specific diseases.
3	This phase includes thousands of patients who have the disease or condition, to confirm clinical efficacy, effectiveness, and safety (confirmatory or pivotal studies).
4	Post-authorization safety studies, real world studies, and registries.

**Table 2 cells-08-00476-t002:** Minimum social and behavioral factors that should be recorded in a patient bibliography.

Race or Ethnic Group, Countries Where People Lived or/and Live (Past and Present), History of Family
Education (years)
Work (type, years, place, and so on)
Economic and social status
Marital status
Social connection or isolation
Stressors (Violence, intimate partner violence, loss of work, loss of a loved one, death in the family or among friends, lack of job, caregiver, and so on)
Diseases (Depression, HIV, and so on)
Physical activity (Days and hours for week engaged in moderate or strenuous exercise)
Tobacco use (Smoked cigarettes per day; and ex-smokers)
Alcohol use (How often and how much alcohol consumed)
Use of prescribed drugs (including HC and HRT), over-the-counter medications, and herbal and nutraceutical use, present and past radiation therapy
Sexual and reproductive history For women: Age of menarche and menstrual history; perimenopause/menopause with associated symptoms; polycystic ovary syndrome; obstetric history (list of all pregnancies and the outcome of each, including abortion; type of anesthesia for delivery (if any); weight of the fetus at delivery; any maternal, fetal, or neonatal complications; and whether the child is currently living should be recorded).

**Table 3 cells-08-00476-t003:** Physiological differences between men and women. Plus (+) indicates a greater extent in a certain sex–gender with respect to the other, or in P (pregnant women) versus NP (non-pregnant women). Minus (–) indicates a lesser extent in a certain sex–gender with respect to the other, or in pregnant women (P) versus non pregnant women (NP). M = men; W = women.

Parameters	P vs NP	M vs W	Comments	References
**Body height**	=	+M		[98]
**Body weight**	+P	+M	Variations in body weight affect drug distribution.	[98,99]
**Body surface area**	+P	+M	Variations in body surface affect drug distribution.	[98,99]
**Fat tissue**	+P	+W	Variations in body composition affect drug distribution. Lipophilic drugs may have a greater volume distribution in women.	[98,99]
**Skeletal muscle**		+M		[98]
**Total body water**	+P	+M	Changes during menstrual cycle.	[98,99]
**Plasma volume**	+P	+M	Changes during menstrual cycle.	[98,99]
**Blood volume**		+M	Even if it is corrected for body weight. Thus, metabolite concentration in blood is diluted more in men than in women, resulting in even greater differences between sexes than if this factor is not considered.	[12,98,99]
**Albumin**	−P			[99]
**Cardiac output**	+P*	−M		[13,95]
**Heart rate**	+P	−M		[10,99]
**QTc interval**		-M		[10]
**Fibrinogen**	+P	−M =	Depends on age, BMI, alcohol consumption in both sexes, and on cigarette smoking, especially in men. Increases with menopause.	[100,101,102]
**Factor II**		−M		[103]
**Factor V**		−M		[103]
**Factor VII**		−M	Increases with menopause.	[100,103]
**Factor VIII**	+P	−M		[102,103]
**Factor X**	+P			[99]
**D-dimer**		−M		[103]
**Fibrinolytic activity**	−P			[102]
**Antithrombin III**	−P	+M, −M	In pre-menopausal women, is lower than men. In post-menopausal, is higher in women than men and decreases with HRT.	[100,102]
**Total protein S (tPS)**	-P	+M, =	In women, increasing age was associated with a significant increase in tPS levels. OC lowered it.	[102,104]
**Free protein S (fPS)**		+M	In women, age had no effect on fPS after adjustment for menopausal state. Not influenced by HRT.	[105]
**Von Willebrand factor**	+P	−M	Increased markedly from non-pregnant values, up to the end of early puerperium.	[103,106]
**Myoglobin**		+M		[107]
**NT-proBNP**		−M	Lower in post-menopausal than premenopausal women and in OC users.	[103,108,109]
**Homoarginine**		+M	Influenced by OC	[109,110]
**Creatinphosphokinase**		+M		[111]
**Total lung capacity**	−P	+M		[99,112,113]
**Residual volume**	−P			[99]
**Tidal volume**	+P			[99]
**Forced vital capacity**		−M		[112]
**Lung function**		+M		[112]
**Airway diameters**		+M		[113]
**Diffusion area**		+M		[113]
**Portal vein flow**	+P			[99]
**Artery hepatic flow**	+		But the increase is not significant	[99]
**Ỵ-glutamyltranspeptidase**		+M		[114]
**Aspartate amino transferase**		+M		[115]
**Alanine aminotransferase**		+M		[115]
**COMT**		+M		[11]
**CYP1A2**	-P	+M	Inducibility is increased (20%) by St. Johns wort in women only.	[11,99]
**CYP2A6**		+W	Increased by OC.	[11]
**CYP3A4**	+P	+W	Inducibility may present SGD. For example, St. Johns wort increases its levels of 50% in men and 90% in women.	[11,99]
**CYP2B6**	+P	+M		[11,99]
**CYP2C9**	+P	=		[11,99]
**CYP2C19**	−P	=	Influenced by OC.	[11,99]
**CYP2D6**	+P	+M		[11,99]
**CYP2E1**	+P	+M		[11,99]
**UGT**	+P	+M		[11,99]
**NAT2**	−P	+M		[11,99]
**Acid secretion**	−P	+M	Men have < absorption of weak acids and > absorption of weak bases; P > absorption of weak bases and < absorption of weak acids.	[11,99]
**Mucus secretion**	+P	+M		[11,99]
**Gastric emptying**	−P	+M		[11,99]
**Intestinal mobility**	−P	+M	P may have a major absorption of drug versus NP; NP may have a major absorption of drug versus men.	[11,99]
**Microbiome**	= (1st trimester); after it changes	Diverge		[86]
**Glomerular filtration rate**	+P	+M	Depends on body weight and serum creatinine levels. If one considers body area, GFR lower is lower by about 10%–25%.	[98,99]
**Renal Blood flow**	+P	+M	When standardized for body surface area.	[98]
**Tubular secretion**	+P	+M	When standardized for body surface area.	[98]
**Tubular reabsorption**	+P	+M	When standardized for body surface area.	[98]
**Creatinine**	−P	+M		[99,116]
**Uric acid**	-P	+M		[117]
**Urea**	−P			[99]
**Cystatin**		+M		[109,118]
**IL-6**		= −W	Influenced by menopause, age, and body weight. Post-menopausal women > exhibit IL-6 responses to acute stress.	[119,120,121]
**IL-18**		+M		[109]
**Tumor necrosis factor-alpha**		−M, +M	Depends on menopausal state, age, and body weight; subcutaneous fat.	[119,120]
**CRP**		−M	CRP appears to be due to a greater accumulation of subcutaneous fat.	[119]
**ICAM**		−M		[103]
**Hsp27**		−M		[103]
**Myeloperoxidase**		−M		[103]
**RAGE**		−M		[103,109]
**Total Cholesterol**		−M	After body weight correction; more elevated in OC users. In absence of body weight normalisation	[120]
				[109]
**HDL**		−M	Increased by OC.	[122]
**LDL**		+M, =		[122]
**Triglycerides**		= +M	In some studies, difference disappears when body composition is considered. Increased by OC.	[122]
**Lp (a)**		−M		[103]
**ApoA-I**		−M		[103]
**ApoC-III**		−M		[103]
**ApoE,**		−M		[103]
**Larger LDL particle size**		−M		[103]
**Leptin**		−M		[103]
**Adiponectin**		-M		[103]
**Resistin**		−M		[103]
**RBC**		+M		[111]
**Haemoglobin**		+M	Fertile age.	[111]
**Hematocrit**		+M		[111]
**Iron**		+M		[111]
**Ferritin**		=, +M	Fertile age.	[111,123]
**Erythropoietin**		+M		[123]
**White blood cells**		=, −M		[120]
**Platelets**		=, −M	Count varies with the menstrual cycle and body weight	[120]
**Malondialdehyde**		−M, =	Fertile women > than men; post-menopausal women = men > 45 years old. After body weight correction, the differences are significant between fertile women and young men; and between post-menopausal women and men > 45 years old.	[120]
**Serum carbonyls**		=		[120]
**Arginine**		+M, =	In men under the age of 45 years; the difference disappeared after correcting for body weight.	[120]
**ADMA**		=, −M	Men < women after body weight correction.	[109,120]
**SDMA**		=, −M	Men < women after body weight correction.	[109,120]
**ADMA/SDMA**		=		[120]
**ADMA/arginine**		=, −M	Men < fertile women.	[120]

ADMA, asymmetric dimethylarginine; APO, apolipoprotein; BMI, body mass index; CRP, C-reactive protein; COMT, catechol-O-methyltransferase; CYP, cytochrome P450 enzymes; HRT, hormone replacement therapy; HDL, high density lipoproteins; Hsp27, heat shock protein 27; ICAM, intercellular adhesion molecule; IL, interleukin; Lp(a), lipoprotein(a); LDL, low density lipoprotein; NAT2, N-acetyltransferase 2; NT-proBNP, N-terminal B-type natriuretic peptide; RAGE, receptor for advanced glycation end products; RBC, red blood cell; SDMA, symmetric dimethylarginine; UGT, uridine diphosphate-glycosyltransferases.

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
