# Peer review of "Sex-Gender Variable: Methodological Recommendations for Increasing Scientific Value of Clinical Studies"

_cells, 2019, doi:10.3390/cells8050476_

Round 1

Reviewer 1 Report

This is a very extensive, interesting and original review of the literature regarding sex and gender differences in research and clinical trials and the development of pharmacological drugs. In addition to this review, the authors propose some very sound and applicable guidelines to implement for future randomized clinical trials.

Major comments:

1 - There are a number of grammatical errors of language which could be improved by revision of the paper by a native English speaker.

2 - The affiliation of Dr Antonini is missing from the list.

3 - Table 1 must be reformated so that the columns are clear and lecture is not confusing.

Minor comments

Table 1: Intestinal motility is more appropriate than intestinal mobility

Table 1: the microbiome row is not clear as the column is spread out.

Author Response

Referee 1

Comments and Suggestions for Authors

This is a very extensive, interesting and original review of the literature regarding sex and gender differences in research and clinical trials and the development of pharmacological drugs. In addition to this review, the authors propose some very sound and applicable guidelines to implement for future randomized clinical trials.

Major comments:

1 - There are a number of grammatical errors of language which could be improved by revision of the paper by a native English speaker. The manuscript has been revised by the MDPI English Editing service.

2 - The affiliation of Dr Antonini is missing from the list. I apologise for the oversight, the affiliations have been corrected

3 - Table 1 must be reformatted so that the columns are clear and lecture is not confusing. Done as requested

Minor comments

Table 1: Intestinal motility is more appropriate than intestinal mobility . the typo has been corrected

Table 1: the microbiome row is not clear as the column is spread out. Done as requested

Reviewer 2 Report

The review is timely and comprehensive. A few major issues that need to be addressed:

·      The abstract needs to be re-written. It is strongly advised to refrain from using colloquial language such as “nonsense”. Please use one “tense” throughout the manuscript.

·      It would be helpful to start with traditional definition of sex, which is based on sex chromosome composition, whereas, gender is assumed or what sex one identifies with. This is an opportunity to set the record straight and not use the terminology sex/gender interchangeably or better define these terminologies. The review refers to this under the subheading “Rules”. Thus, to be consistent with their own suggestion, the introduction should be revised.

·      Reference 10 is a review article. It is suggested to include original article to give due credit.

·      Lines 40-43: Example of thalidomide is important, but as presented a bit misleading. The drug was not tested in non-rodent mammals. Please see this reference for a historical perspective: PMID: 26043938 and revise accordingly.

·      Line 48: “males are considered as proxy for women”. Not sure if this is true. More commonly, males of the species are used due to convenience and “apparent lack” of hormonal changes.

·      Line 63: RCT: please define abbreviations when they first appear in text. It is suggested not to abbreviate unless it appears more than 5 times in the text.

·      Section 2.2, Rule 2. It might be helpful to suggest that articles and manuscript precisely describe the sex of the researcher or physician? This may help explain differences in findings, despite identical protocols being used.

·      It might be helpful to include a separate section on cultural differences in medical care or perceptions.

·      Section 2.8, rule 8. Are the authors advocating for inclusion of pregnant and lactating mothers in RCTs? Citing the example of thalidomide use, ethical and long-term adverse effects on the fetus and the progeny raises several ethical issues. Perhaps focusing on FDA regulations regarding inclusion and exclusion criterion might be helpful. Empirical evidence must be provided and benefits should outweigh the risk.

·      Overall, the manuscript needs to be edited for grammar and language use.

Author Response

Referee 2

Comments and Suggestions for Authors

The review is timely and comprehensive. A few major issues that need to be addressed:

·      The abstract needs to be re-written. It is strongly advised to refrain from using colloquial language such as “nonsense”. Please use one “tense” throughout the manuscript. The manuscript has been revised by the MDPI English Editing service. The abstract has been modified

·      It would be helpful to start with traditional definition of sex, which is based on sex chromosome composition, whereas, gender is assumed or what sex one identifies with. This is an opportunity to set the record straight and not use the terminology sex/gender interchangeably or better define these terminologies. The review refers to this under the subheading “Rules”. Thus, to be consistent with their own suggestion, the introduction should be revised. I thank the referee for her/his suggestion, the terminology has been clarify, adding also some references and the introduction has been modified

·      Reference 10 is a review article. It is suggested to include original article to give due credit. The reference 10 has been replaced as requested

·      Lines 40-43: Example of thalidomide is important, but as presented a bit misleading. The drug was not tested in non-rodent mammals. Please see this reference for a historical perspective: PMID: 26043938 and revise accordingly. I thank the referee for her/his suggestion. The lines has been re-written and new references have been added accordingly.

·      Line 48: “males are considered as proxy for women”. Not sure if this is true. More commonly, males of the species are used due to convenience and “apparent lack” of hormonal changes. I agree with the referee, and the sentence has been clarify

·      Line 63: RCT: please define abbreviations when they first appear in text. It is suggested not to abbreviate unless it appears more than 5 times in the text. The abbreviation was defined at the first appearance in the text

·      Section 2.2, Rule 2. It might be helpful to suggest that articles and manuscript precisely describe the sex of the researcher or physician? This may help explain differences in findings, despite identical protocols being used.    It might be helpful to include a separate section on cultural differences in medical care or perceptions. As requested, the rule has been improved adding more information.

·      Section 2.8, rule 8. Are the authors advocating for inclusion of pregnant and lactating mothers in RCTs? Citing the example of thalidomide use, ethical and long-term adverse effects on the fetus and the progeny raises several ethical issues. Perhaps focusing on FDA regulations regarding inclusion and exclusion criterion might be helpful. Empirical evidence must be provided and benefits should outweigh the risk. As requested, FDA regulations have been taken into consideration. it is important to carry out clinical studies during pregnancy and lactation paying attention to the potential fetal/infant risks

·      Overall, the manuscript needs to be edited for grammar and language use. The manuscript has been revised by the MDPI English Editing service.

Round 2

Reviewer 2 Report

Suggested changes.

 Lines 12-13: Attitudes concerning the enrollment of women in randomized clinical trials has changed over recent years. Although Despite this change, a gap is still present. 

Lines 21-22: Thus, This review proposes a set of 12 rules to improve clinical research for integrating sex-gender into clinical trials. 

Lines 28-32: The text is duplicated. Please consider consolidating. Maybe revise to:

While sex and gender are often used interchangeably, it is important to point out that these terms should not be used as synonyms, especially when designing clinical trials. In general, sex refers to the biological, genetic (two X chromosomes (XX), as opposed to XY), and physiological processes related to sexual beings [1-4]. Whereas gender refers to the roles, relationships, and relative power that people relate to or societies generally attribute to women, men, irrespective of their genetic make-up. 

Lines 47-55: Please remove duplication and rephrase.

Lines 60-61: male subjects are mainly used in research. with the results then applied also to the female gender . Results are often extrapolated and thought to be applicable to women, which may not be true.

Author Response

I thank the referre for his / her suggestions. As requested the senteces have been modified, changing words and removing duplicates.